

# Nonexistence of motility induced phase separation transition in one dimension

**Indranil Mukherjee, Adarsh Raghu and Pradeep Kumar Mohanty**⋆

Department of Physical Sciences,
Indian Institute of Science Education and Research Kolkata, Mohanpur, 741246 India

⋆ pkmohanty@iiserkol.ac.in

## Abstract

We introduce and study a model of hardcore particles obeying run-and-tumble dynamics on a one-dimensional lattice, where particles run in either +ve or −ve $x$-direction with an effective speed $v$ and tumble (change their direction of motion) with a constant rate $\omega$ when assisted by another particle from right. We show that the coarse-grained dynamics of the system can be mapped to a beads-in-urn model called misanthrope process where particles are identified as urns and vacancies as beads that hop to a neighbouring urn situated in the direction opposite to the current. The hop rate is same as the magnitude of the particle current; we calculate it analytically for a two-particle system and show that it does not satisfy the criteria required for a phase separation transition. Nonexistence of phase separation in this model, where tumbling dynamics is rather restricted, necessarily imply that motility induced phase separation transition can not occur in other models in one dimension with unconditional tumbling.



# 1 Introduction

An important class of nonequilibrium systems is that of active matter systems (AMS) [1] where the individual constituents are self-propelled; instances of such systems include bird flocks [2], bacterial colonies [3], photophoretic colloidal suspensions [4] and actin filaments [5] etc. They exhibit a number or interesting features like large number-fluctuations [1], clustering and pattern formation [4]. A major area of interest in the study of AMS has been the so-called *motility-induced phase separation* (MIPS) [6–13] which refers to spatially separated high and low density regimes. Such aggregation or clustering of particles has been observed experimentally in many active matter systems [4]. Relevance of the aggregation process has also been proposed as a mechanism of formation bacterial biofilms [3], which are sources of infection.

Occurrence of MIPS relies on an argument that effective velocity of active particles decrease in crowded or high density regions formed either by explicit dependence of local density or merely by exclusion. Naturally such a slowing down of movement further increases the density of particles and gives rise to a feedback loop allowing the stable high density (liquid-like) regions to form and coexist with a low density (gas-like) phase elsewhere. MIPS has been widely investigated in simulations and apparent phase separation has been observed. Theoretical investigations of this phenomenon have thus far concentrated on continuum models [8–10] where motility parameters, such as particle flux or velocity are characterized as functions of the coarse-grained local density [6, 7]. Lattice models of active particles have been studied in one and two dimensions numerically [14–16] with run and tumble particles (RTPs). RTPs move at a fixed speed along the direction of their orientation (a *run*) until they *tumble* and change their orientation. In one dimension (1D), the two orientations (say, $\pm$) are usually referred to as the internal degrees of the particle (*spin*), which flips with a certain rate. Analytical studies of these lattice models are limited. Thompson et. al. [13] have introduced a model of self propelled particles with RTP dynamics; in 1D. These models exhibit inhomogeneous density profiles when particle velocities depend on their position. Recently Slowman et. al. [17, 18] have obtained an exact solution for *two* RTPs and found jamming induced attraction between the particles of the opposite spins, which indicates that, for many particle systems, a phase separated state might originate from these attractive interactions. Later, Dandekar et. al. [19] have obtained a mean-field solution of RTPs in 1D which turned out to be a good approximation when tumbling rate is large.

An element of surprise in the formation of a phase separated state *without* any explicit attractive interaction has generated much excitement to the study of MIPS and raised questions about the stability of such states in 1D in absence of any explicit interaction or spatial potential. Recent works have added to the doubt by showing that MIPS phase transition in 2D belongs to the Ising universality class [20–22] which does not have a counterpart in one dimension. In this article we argue and show explicitly using 1D lattice models of RTPs that indeed MIPS transition can not occur in 1D; the inhomogeneous states observed in numerical simulations and in hydrodynamic models are only long lived transient states.

First we introduce a generic model of hardcore RTPs in 1D with a restricted tumbling dynamics and show that its coarse-grained dynamics can be mapped to a beads-in-urn model, namely a misanthrope process [23, 24] where beads hop to their neighbouring urn, situated in the opposite direction of the particle current, with a rate same as the magnitude of current. The functional form of hop rate is determined from the exact steady state results of the model with only two RTPs. To determine if MIPS transition is possible, we use the following criterion. If a system of hardcore particles phase separates as its density $\rho$ crosses a threshold $\rho^*$ then the maximum density at which it remains homogeneous is $\rho^*$. Since systems with homogeneous densities are well described in the grand canonical ensemble (GCE) by a unique chemical potential $\mu$ (or fugacity $z = e^\mu$), we argue that phase separation transition is possible in a

system when its density in GCE attains a maximum value $\rho^* = \text{Max}[\rho(z)]$ which is less than unity (the density of a fully occupied lattice). Nonexistence of MIPS transition in restricted tumbling model would imply that MIPS can not occur in any other RTP model in 1D where tumbling occurs more frequently.

## 2 The restricted tumbling model

We introduce a generic model of RTPs on an one dimensional periodic lattice with sites labeled by $i = 1, 2, \dots L$. The sites are either empty (represented by $\tau_i = 0$) or occupied by at most one RTP $\tau_i = \pm$ having orientation (spin) $\pm$. Particles follow a run dynamics,

$$+0 \underset{q_+}{\overset{p_+}{\rightleftharpoons}} 0+ \, ; \qquad -0 \underset{q_-}{\overset{p_-}{\rightleftharpoons}} 0- \, , \tag{1}$$

where RTPs move forward or backward with rates $p_\pm$ and $q_\pm$ respectively. Along with this, they can tumble and change their spin with rate $\omega$ as follows,

$$+\pm \overset{\omega}{\rightarrow} -\pm \, ; \qquad -\pm \overset{\omega}{\rightarrow} +\pm \, . \tag{2}$$

Tumbling is restricted here in the sense that *only* those particles which are assisted from right by other particles can tumble their direction. This restriction helps us getting an approximate steady state of the system without tampering the main aim: the proposition that a stable MIPS state *can not* be sustained in 1D. Since frequent tumbling of particles helps the system to clear jamming, a proof of nonexistence of MIPS in our model necessarily guarantees its nonexistence in any other model that has more liberal tumbling dynamics. Hereafter we refer to the model following dynamics (1) and (2) as restricted tumbling model (RTM).

Although RTM is defined for generic rates $(p_\pm, q_\pm)$ we study the case $p_\pm = q_\mp$ where the run dynamics exhibit a symmetry transformation, namely simultaneous interchange of parity (left $\rightleftharpoons$ right) and spin ($+ \rightleftharpoons -$), that keeps the dynamics invariant. This symmetry was present for both run- and tumble-dynamics in 1D lattice models studied earlier [17, 19]. When $p_\pm = q_\mp$, it is also ensured that in the limit when lattice spacing vanishes [25], a single particle dynamics of RTM reduces to that of a RTP moving in continuum space with same speed $v = p_- - q_- = q_+ - p_+$ along +ve and −ve $x$-directions. Note that, under parity transformation (left $\rightleftharpoons$ right) the tumbling dynamics of our model is modified as tumbling now occurs for only those particles which are assisted by other particles from left. But, for $p_\pm = q_\mp$, a left-assisted tumbling dynamics leads to the same steady state as the right-assisted tumbling. This can be verified easily from the exact mapping of these models to the corresponding beads-in-urn models (see later discussions).

A special case of RTM with $p_+ = \alpha = q_-, p_- = 0 = q_+$ and unrestricted tumbling dynamics $+ \underset{\omega}{\overset{\omega}{\rightleftharpoons}} -$ was studied earlier by Slowman et. al. [17] and an exact steady state solution was obtained for a system of two RTPs. It turned out that these two particles experience an effective attractive interaction in the steady state when their spins are opposite; it is envisaged that this attraction might be the source of MIPS states observed in corresponding hydrodynamic models. In comparison, in Eq. (2) we have dropped one of the transition $+0 \underset{\omega}{\overset{\omega}{\rightleftharpoons}} -0$; as a consequence, particles do not tumble if they are not assisted by a right neighbour.

## 3 Mapping of RTM to a beads-in-urn model

Any microscopic configurations $\{\tau_i\}$ of RTM can be viewed as urns containing beads –each particle is an urn that contains beads which are uninterrupted sequence of 0s (vacancies) to

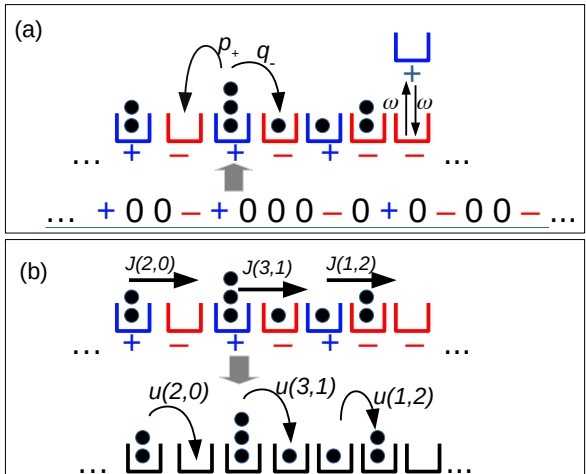

Figure 1: (a) Mapping lattice model of RTPs to an urn model. (b) Effective coarse-grained dynamics: hop rate of a bead $u(m_k, m_{k+1})$ from urn $k$ (with $m_k$ particles) to $k+1$ (with $m_{k+1}$ particles) is assumed to be same as the local bead current $J(m_k, m_{k+1})$ averaged over internal degrees $\sigma_k, \sigma_{k+1}$.

the right of the particle (as described in Fig. 1(a)). The spin $\pm$ of the particle is termed as the internal degree of the urn. Thus we have a beads-in-urn model of $N$ urns indexed by $k = 1, 2, \ldots N$, each carrying an internal degree $\sigma_k = \pm$ and $m_k = 0, 1, 2 \ldots$ beads. The dynamics (1) and (2) now translate to hopping of a bead from urn $k$ to $k+1$ ($k-1$) with rate $q_{\sigma_{k+1}}$ ($p_{\sigma_k}$), and flipping of internal degrees $\sigma_k \to -\sigma_k$ with rate $\omega \delta_{m_k, 0}$. The total number of beads $\sum_{k=1}^{N} m_k = L - N \equiv M$ is conserved by the dynamics. Like particle density $\rho = \frac{N}{L}$, the bead density $\eta = \frac{M}{N} = \frac{1-\rho}{\rho}$ is also conserved.

Note that in this beads-in-urn model the internal degrees of the urns can flip only when they are empty; this restriction forces $k$-th urn either to transfer a bead (when $m_k > 0$) *or* to change the internal degrees (when $m_k = 0$) and help us getting an exact steady state. It is easy to see that a left-assisted tumbling dynamics with same rate $\omega$ will also map to the *same* beads-in-urn dynamics when particles are identified as urns containing number of beads same as the consecutive vacancies to their left and the hope rates are $p_\pm = q_\mp$.

The mapping of RTM to beads-in-urn model is exact but its steady-state could not be obtained analytically. We proceed to develop a coarse-grained picture. In the steady state of the urn model, the local bead current $J$ (summed over $\pm$ degrees) effectively transports the beads from one urn to its neighbour situated along the direction of total current. Since hop-rates ($q_{\sigma_{k+1}}, p_{\sigma_k}$) in the original beads-in-urn model were dependent on spins of neighbouring urns it is expected that the local bead current must depend on the number of beads present in neighbouring urns, i.e. $J \equiv J(m_k, m_{k+1})$. This current can be set as the effective hop-rate of a coarse-grained model where urns lose their internal degrees and a single bead hops from urn $k$ to $(k+1)$ with rate $u(m_k, m_{k+1}) = J(m_k, m_{k+1})$; rightward hopping ($k$ to $(k+1)$) is considered assuming that the current is flowing in +ve $x$-direction. Thus, in this coarse-grained picture (see Fig 1(b)), all urns are equivalent (as they lose their internal degrees) and the hop-rate depends on the number of beads present in the departure and the arrival urn; such a process is called a misanthrope process (MAP) [23, 24].

In fact, mapping of hardcore particle systems to urn model with an exact or effective coarse-grained dynamics, similar to the dynamics of a *zero range process* (ZRP) [26] are quite reliable and have helped researchers [27] earlier to establish non-existence of phase separation transition in certain lattice models [28] where rigorous numerical simulations have exhibited apparent phase separated states. It also helped in predicting *true* phase separation transition in many other models [27, 29, 30]. In contrast, mapping to that of misanthrope process, that we introduce here, provides a better coarse-grained picture as steady-state correlation between neighbouring urns are retained here.

The bead-current $J(m_k, m_{k+1})$ flowing across the urns can be computed from numerical simulations (will be discussed later), but that does not help us to compute $\rho(z)$ in grand canonical ensemble. To calculate $\rho(z)$ we need functional form of $J(m_1, m_2)$ which can be calculated exactly using matrix product ansatz (MPA) [31] for a system of two urns containing $M$ number of beads (i.e., $L = M + 2$), each one following the dynamics described in Fig. 1(a).

For urn models, a matrix product steady state (MPSS) can be obtained following Ref. [32]. We now consider RTM model, which is mapped exactly to the urn model described in Fig. 1(a). The steady state probability of a generic configuration $\{\sigma_k m_k\}$, where $k^{th}$ urn (spin $\sigma_k$) has $m_k$ beads, is given by a matrix product ansatz,

$$P(\{\sigma_k m_k\}) \sim Tr\left[\prod_{k=1}^{N} X_{\sigma_k}(m_k)\right]\delta\left(\sum_{k=1}^{N} m_k - M\right),\tag{3}$$

where matrix $X_{\sigma_k}(m_k)$ represents the $k^{th}$ urn having internal degree $\sigma_k$ and $m_k$ beads. The $\delta$-function here ensures that the total number of beads $M$ are conserved. These matrices are constrained to follow a matrix algebra so that $P(\{\sigma_k m_k\})$ defined above must satisfy the steady state condition $\frac{dP}{dt} = 0$ for the dynamics in Fig 1(a). We find (see Appendix) that for $N = 2$, matrices $X_\sigma(m)$ have a $2 \times 2$ representation (for any $\omega > 0$),

$$X_+(m) = \begin{bmatrix} 1 & 0 \\ 1 & 0 \end{bmatrix}, \qquad X_-(m) = \gamma^m \begin{bmatrix} 0 & 1 \\ 0 & 1 \end{bmatrix}; \qquad \gamma = \frac{p_+ + q_-}{p_- + q_+}.\tag{4}$$

The steady state probabilities of two urns containing $m_1, m_2$ beads are then, $P_{\sigma_1 \sigma_2}(m_1, m_2) = \frac{1}{Q_M} Tr[X_{\sigma_1}(m_1)X_{\sigma_2}(m_2)]\delta(m_1 + m_2 - M)$, where $Q_M = \sum_{\sigma_1,\sigma_2} \sum_{m_1=0}^{M} Tr[X_{\sigma_1}(m_1)X_{\sigma_2}(M-m_1)]$. Explicitly,

$$P_{\sigma_1 \sigma_2}(m_1, m_2) = \frac{1}{Q_M}\gamma^{\frac{1}{2}(1-\sigma_1)m_1 + \frac{1}{2}(1-\sigma_2)m_2},\tag{5}$$

with $m_2 = M - m_1$. Thus, the average local current carried by the beads when the two urns have $(m_1, m_2)$ particles is

$$J(m_1, m_2) = \sum_{\sigma_1,\sigma_2} P_{\sigma_1 \sigma_2}(m_1, m_2)(q_{\sigma_1} - p_{\sigma_2})$$
$$= \frac{1}{Q_{m_1+m_2}}\left[(q_+ - p_+) + (q_+ - p_-)\gamma^{m_1} + (q_- - p_+)\gamma^{m_2} + (q_- - p_-)\gamma^{m_1+m_2}\right].$$

For RTPs, which need to satisfy the condition $p_\pm = q_\mp$,

$$J(m_1, m_2) = v\frac{1 - \gamma^{m_1+m_2}}{Q_{m_1+m_2}},\tag{6}$$

where $v = p_- - q_- = q_+ - p_+$ and $\gamma = \frac{p_+}{p_-}$ (as in Eq. (4)). Note that $J(m_1, m_2)$ depends only on the sum of its arguments, i.e., $J(m_1, m_2) \equiv J(m_1 + m_2)$. We will now set $J(m_1 + m_2)$ as

the hop-rate of beads in the coarse-grained model, i. e., $u(m_k, m_{k+1}) = J(m_k + m_{k+1})$. This urn model is a misanthrope process where hop-rate is a function of total number of beads present in the departure and the arrival site. It turns out that the steady state of this specific misanthrope process has a factorized form,

$$P(\{m_k\}) \sim \prod_{k=1}^{N} f(m_k), \quad \text{with} \quad f(m) = \prod_{n=1}^{m} \frac{u(1, n-1)}{u(n, 0)} = 1.$$

The grand partition function with a fugacity $y$ that controls the total number of beads $M \sum_{k=1}^{N} m_k$ is

$$\mathcal{Q}_N(y) = \sum_{\{m_k\}} P(\{m_k\}) y^{m_k} = F(y)^N ;$$
$$F(y) = \sum_m f(m) y^m = \frac{1}{1-y}. \tag{7}$$

In RTM, both $N, M = \sum_{k=1}^{N} m_k$ vary keeping the system size $L$ fixed. To account for that we introduce another fugacity $z$, so that the new partition function is,

$$Z(z, y) = \sum_{N=0}^{\infty} \mathcal{Q}_N(y) z^N = \frac{1}{1 - zF(y)}, \tag{8}$$

which gives rise to $\langle N \rangle = z \frac{\partial}{\partial z} \ln Z(z, y)$ and $\langle M \rangle = y \frac{\partial}{\partial y} \ln Z(z, y)$. We now set $\langle N \rangle + \langle M \rangle \equiv L$ to obtain $z$ in terms of $y$, $z = \frac{L}{(1+L)F(y) + yF'(y)}$. Then,

$$\rho(y) \equiv \frac{\langle N \rangle}{L} = \frac{F(y)}{F(y) + yF'(y)} = 1 - y. \tag{9}$$

The maximum value of the RTP density, obtained when $y \to 0$, $\rho^* = 1$. Thus the fugacity $y$ can *always* be tuned to obtain any arbitrary particle density $0 \leq \rho \leq 1$; thus, irrespective of the value of $\rho$, the system remains homogeneous and *can not* phase separate.

The above argument is based on a coarse-grained picture where the hop rate $u(m, n) \equiv u(m + n)$ is taken same as the average local current of beads. In the following we employ a method to calculate $J(.)$ numerically from Monte Carlo simulations of the model and compare it with Eq. (6).

To simulate the dynamics we must set $p_\pm = q_\mp$ required for the system to have a valid RTP dynamics, which gives $\gamma = \frac{p_+}{p_-}$ in Eq. 4. Without loss of generality we can set $p_- = 1 = q_+$, by choosing a suitable time unit; then, $p_+ = q_- = \gamma$ and the speed of RTPs $v = q_+ - p_+ = 1 - \gamma$. We also consider $\gamma \leq 1$ ($\gamma > 1$ case can be explored directly by using left/right and $+/-$ symmetry). From Eq. (6), $J(m) = \frac{v}{Q_m}(1 - \gamma^m)$, which has an asymptotic form (for large $m$),

$$J(m) \equiv u(m) \simeq m \frac{1-\gamma}{m+c}; \qquad c = \frac{3-\gamma}{1-\gamma}. \tag{10}$$

This implies that $u(m)^{-1}$ is a linear function of $m^{-1}$ with slope $c(1-\gamma)^{-1}$ and $y$-intercept $(1-\gamma)^{-1}$, which we verify from the Monte Carlo simulations of the urn model (Fig 1(a)). For a given value of $\gamma, \rho, \omega$ first we allow the system to relax for a long time starting from a random initial configuration. The system may take a very long time to reach a true phase separated state when it exists, but the hoping dynamics in the coarsening regime given by $u(m_1, m_2) = J(m_1, m_2)$ can predict, well in advance, if the system is approaching towards a inhomogeneous (MIPS) or a homogeneous state.

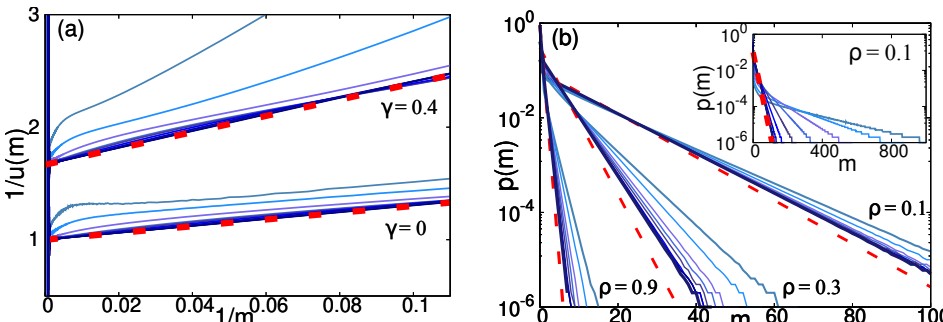

Figure 2: Simulation of RTM model with dynamics (1) and (2) (equivalently an urn model described in Fig. 1(a)). (a) Hop rate $u(m)^{-1}$ obtained from numerical simulations (solid line) for $\gamma = 0, 0.4$ and $\omega = 0.005$ to 1 (top to bottom) are compared with Eq. (10) (dashed line) when $\rho = 0.02$. All the curves approach linearly to the asymptotic value $(1-\gamma)^{-1}$, as predicted. (b) Marginal distribution $p(m)$ of the separation $m$ are compared for $\gamma = 0$ and $\rho = 0.1, 0.3, 0.9$ in semi-log scale. Solid lines (results from simulations for $\omega = 0.2$ to 10 (right to left) are shown along with dashed lines, $\rho(y)^m$ with $y = 1-\rho$ obtained from coarse-grained description of the model. The inset shows the same for $\rho = 0.1$ but smaller $\omega = 0.005$ to 1 (right to left). In all cases $p(m)$ shows exponential behaviour; but for small $\omega$, $y$ differs substantially from the predicted value $(1-\rho)$. Here, $p_+ = \gamma = q_-$, $p_- = 1 = q_+$, $L = 10^4$. In each case, statistical averaging is done for more than $10^7$ samples.

In the coarsening regime we consider a large time interval and calculate $(F_r(m_1 + m_2), F_l(m_1 + m_2))$, the number of times beads move to (right, left) when the departure and arrival urns have exactly $m_1$ and $m_2$ beads respectively (internal degree of the urns are ignored). Also, we keep track of $F(m_1 + m_2)$, the number of jump-events attempted during that interval. Clearly, $u(m) = (F_r(m) - F_l(m))/F(m)$. In Fig. 2(a) we plot $u(m)^{-1}$ versus $m^{-1}$ for $\gamma = 0, 0.4, \rho = 0.02$ and $\omega = 0.005$ to 1; in all cases, $u(m)^{-1}$ is found to be linear for large $m$ as expected from Eq. (10). The $y$- intercepts also approach to the known value $(1-\gamma)^{-1}$ but the slopes differ a bit. Further, in Fig. 2(b) we plot the marginal distribution $p(m)$ of number beads $m$ for $\gamma = 0, \rho = 0.1, 0.3, 0.9, \omega = 0.2$ to 10. The dashed line corresponds to the theoretical curve obtained from the coarse-grained picture: $p(m) = y^m f(m)/F(y) = \rho y^m$ where $y = 1-\rho$. In all cases, as shown Fig. 2(b), $p(m)$ exhibits exponential distributions that match very well with the prediction when $\omega$ is large. As $\omega \to 0$ the exponential feature remains persistent but the value of $y$ differs substantially from the theoretical value $1-\rho$. This is because ergodicity is broken at $\omega = 0$; the system there falls into one of the fully jammed (or absorbing) configuration and remains there.

Essentially, the coarse-grained picture turns out to be a good description of the RTP model as $p(m)$ decays exponentially for large $m$ as predicted - rest of the details are less relevant because an exponential form of $p(m)$ is enough to assure that the fugacity in GCE can *always* be tuned to secure any desired particle density $0 < \rho < 1$. Such a system can not support any stable MIPS phase and settles to form a homogeneous density profile for all $\omega > 0, \gamma \geq 0$.

The above conclusion can also be obtained from using an approximate matrix product steady state (MPSS). Matrix representations (4), that provides exact MPSS exclusively for $N = 2$, are also excellent approximations for larger $N$ (justified in the Appendix). With these matrices, for $N > 2$, the grand partition function $Z(z, y)$ and density $\rho(y)$ are given by Eqs.

(A.6) and (A.7) respectively,

$$Z(z,y) = \frac{1}{1 - zF(y)}; \; F(y) = \frac{1}{1-y} + \frac{1}{1-\gamma y}$$
$$\text{and} \quad \rho(y) = \frac{(1-y)(1-\gamma y)(2-y-\gamma y)}{(1-\gamma y)^2 + (1-y)^2}. \tag{11}$$

Clearly, the maximum density that can be achieved in GCE by tuning $y$ is $\rho^* = 1$ (when $y = 0$) and thus, this RTP model *can not* undergo a phase separation transition at any $\rho < 1$. One can safely extend these results for restricted tumbling dynamics to other RTP models where tumbling occurs more frequently; this is because tumbling is generally detrimental to the stability of MIPS. Our conclusions are consistent with the recent results [20–22] that MIPS transition in 2D belongs to the Ising universality class that does not have an one dimensional analogue.

## 4 Summary

In this article we show that phase separation of *free* hardcore-RTPs with constant run and tumble rates is not possible in 1D. One may however add some crucial features which are known to enhance or freshly produce phase separated states of passive particles, like invoking explicit attractive interaction [27] *or* making tumbling rates to decrease with $L$ (so that it vanishes in the thermodynamic limit) [33] *or* explicitly forcing the run dynamics to depend on (and reduce substantially with increase of) local particle density [26] *or* adding impurities [34]. Then a phase separation transition may occur, but will it keep its charm and glory to be identified as the *motility induced* phase separation, particulary when the transition is anyway expected for similar system of passive particles (without motility)?

Recently Kourbane-Houssene et. al. [35] have introduced a RTP model where the difference of run-rates (or effective velocity) are taken proportional to $\frac{1}{L}$ and the tumbling rate is proportional to $\frac{1}{L^2}$ (downplayed by a factor $1/L$ compared to the run rates); using an exact coarse-grained hydrodynamic description they show that a homogeneous phase in 1D loses its stability in certain parameter regimes. Another way might be to use strongly biased tumbling rates where, say, $+ \rightarrow -$ occurs much more frequently than $- \rightarrow +$. In this case a phase separation transition occurs [33] when $q_\pm = 0$, where the dynamics of RTM reduces to that of a two species exclusion process [36]. Its extension to small $q_\pm \simeq 0$, is a RTP model (having a good continuum limit) and it is reasonable to assume that the phase separation features may also survive there. Yet another possibility is to introduce defects. Recent studies [37] have shown that a jammed phase does exist in RTM like models with defects. More investigations are required in all these directions to confirm if RTP models in 1D can phase separate.

## Acknowledgements

PKM acknowledges stimulating discussions with Urna Basu. IM acknowledges the support of Council of Scientific and Industrial Research, India (Research Fellowship, Grant No. 09/921(0335)/2019-EMR-I).

# A  Appendix

The dynamics (1) and (2) of RTM can be mapped exactly to an urn model described in Fig. 1(a) where beads hop from site $k$ to site $k+1$ (or site $k-1$) with rates $q_{\sigma_{k+1}}$ (or $p_{\sigma_k}$) respectively. The probability density of a generic configuration $\{\sigma_k m_k\}$ evolves following the Master equation,

$$
\begin{aligned}
\frac{d}{dt}&P(\ldots,\sigma_{k-1}m_{k-1},\sigma_k m_k,\sigma_{k+1}m_{k+1},\ldots)\\
&=-(p_{\sigma_k}+q_{\sigma_{k+1}})P(\ldots,\sigma_{k-1}m_{k-1},\sigma_k m_k,\sigma_{k+1}m_{k+1},\ldots)\\
&\quad+q_{\sigma_k}P(\ldots,\sigma_{k-1}m_{k-1}+1,\sigma_k m_k-1,\sigma_{k+1}m_{k+1},\ldots)\\
&\quad+p_{\sigma_{k+1}}P(\ldots,\sigma_{k-1}m_{k-1},\sigma_k m_k-1,\sigma_{k+1}m_{k+1}+1,\ldots)\\
&\quad-\omega\delta_{m_k,0}P(\ldots,\sigma_{k-1}m_{k-1},\sigma_k m_k,\sigma_{k+1}m_{k+1},\ldots)\\
&\quad+\omega\delta_{m_k,0}P(\ldots,\sigma_{k-1}m_{k-1},-\sigma_k m_k,\sigma_{k+1}m_{k+1},\ldots),
\end{aligned}
\tag{A.1}
$$

where first three terms in the right hand side corresponds to the run dynamics and the rest describes tumbling at a generic site $k$. In the steady state $\frac{d}{dt}P(\{\sigma_k m_k\})$ must vanish; this, along with the matrix product ansatz (3) leads to $\sum_{k=1}^N \mathrm{Tr}[H_k^R + H_k^T] = 0$, where $H_k^R$ and $H_k^T$ correspond to the run and the tumble dynamics respectively,

$$
\begin{aligned}
H_k^R =& -(p_{\sigma_k}+q_{\sigma_{k+1}})X_{\sigma_{k-1}}(m_{k-1})X_{\sigma_k}(m_k)X_{\sigma_{k+1}}(m_{k+1})\\
&+q_{\sigma_k}X_{\sigma_{k-1}}(m_{k-1}+1)X_{\sigma_k}(m_k-1)X_{\sigma_{k+1}}(m_{k+1})\\
&+p_{\sigma_{k+1}}X_{\sigma_{k-1}}(m_{k-1})X_{\sigma_k}(m_k-1)X_{\sigma_{k+1}}(m_{k+1}+1)
\end{aligned}
\tag{A.2}
$$

$$
\text{and} \quad H_k^T = \omega[X_{-\sigma_k}(0)-X_{\sigma_k}(0)]X_{\sigma_{k+1}}(m_{k+1}).
$$

We now introduce some suitable choice of auxiliary matrices $\tilde{X}_{\sigma_k,\sigma_{k+1}}(m_k,m_{k+1})$, yet to be determined along with $X_{\sigma_k}(m_k)$, so that both $\sum_k H_k^R$ and $\sum_k H_k^T$ vanish separately; one such cancellation scheme for $H_k^R$ is,

$$
H_k^R = \tilde{X}_{\sigma_{k-1},\sigma_k}(m_{k-1},m_k)X_{\sigma_{k+1}}(m_{k+1}) - X_{\sigma_{k-1}}(m_{k-1})\tilde{X}_{\sigma_k,\sigma_{k+1}}(m_k,m_{k+1}).
\tag{A.3}
$$

We find that a choice $\tilde{X}_{\sigma,\sigma'}(m,n) = h_{\sigma\sigma'}X_\sigma(m)X_{\sigma'}(n)$ with some scalar parameter $h_{\sigma\sigma'}$ does satisfy the steady state condition with $2\times 2$ matrices

$$
X_+(m)=\begin{bmatrix}1&0\\1&0\end{bmatrix}, \qquad X_-(m)=\gamma^m\begin{bmatrix}0&1\\0&1\end{bmatrix},
\tag{A.4}
$$

when $\gamma=\frac{p_++q_-}{p_-+q_+}$, $h_{+-}=0=h_{-+}$ and

$$
h_{++}=h_{--}=\begin{cases} q_\sigma(1-\gamma^\sigma), & m>0,\ n>0,\\ 0, & \text{else.}\end{cases}
\tag{A.5}
$$

These matrices also satisfy the condition $\sum_k \mathrm{Tr}[H_k^T]=0$ set by the tumbling dynamics because $X_\sigma(0)X_{\sigma'}(m)=X_{\sigma'}(m)$ for all $\sigma,\sigma',m$. The only troubling part is that $h_{\sigma\sigma}$s depend implicitly on $m,n$ violating the assumption that they are constants. This implicit dependence of $h_{++}$ and $h_{--}$ on $m,n$ drops out when (i) $q_\pm=0$ (all particles move in the same direction), (ii) $\gamma=1$ (which sets the speed of RTPs $v=1-\gamma=0$ when $p_\pm=q_\mp$). In both cases we have an exact MPSS, but neither of these cases constitutes the scenario of MIPS. Yet another case is $N=2$ where matrices given by Eq. (A.4) leads to an exact MPSS. This is because the cancellation scheme in Eq. (A.2) acts on product of *three* consecutive matrices which are not present when $N=2$; thus, one can make $h_{\sigma\sigma'}$ independent of $m,n$ by setting safely $h_{\sigma\sigma'}=0$ for all $\sigma,\sigma'$. Steady state probabilities for $N=2$ is given by Eq. (5).

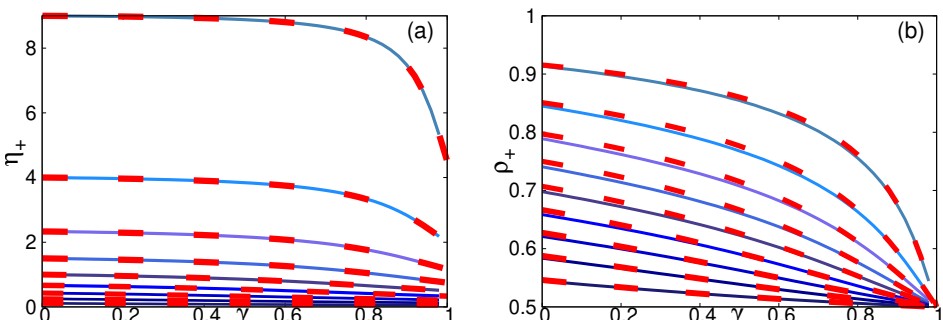

Figure 3: (a) $\eta_+$, the density of beads in $+$ urn and (b) $\rho_+$, the fraction of $+$ urns are shown as a function of $\gamma$ for different $\rho = 0.1$ to $0.9$ (top to bottom). Data from Monte Carlo simulations (solid lines) of RTM model described in Fig. 1(a), averaged over $10^7$ samples are compared with Eqs. (A.9) (dashed line). Other parameters are $L = 10^3$, $p_+ = \gamma = q_-$, $p_- = 1 = q_+$ and $\omega = 1$.

Now we proceed for larger $N$ and get an approximate MPSS while dependence of $h_{\sigma\sigma'}$ on $m, n$ are ignored and both $h_{++}$ and $h_{--}$ are taken as $q_\sigma(1 - \gamma^\sigma)$ $\forall m, n \geq 0$. We will see that the matrices (A.4) provide a MPSS which are an excellent approximation to the exact ones. The canonical partition function of the system is

$$Q_{M,N} = \sum_{\{\sigma_k m_k\}} Tr\left[\prod_{k=1}^{N} X_{\sigma_k}(m_k)\right] \delta\left(\sum_{k=1}^{N} m_k - M\right),$$

and the grand partition function, with fugacities $z, y$ associated with $N, M$, is

$$Z(z, y) = \sum_{M=0, N=0}^{\infty} z^N y^M Q_{M,N} = \sum_{N=0}^{\infty} z^N F(y)^N,$$

$$F(y) = \sum_{\sigma=\pm} \sum_{m=0}^{\infty} y^m Tr[X_\sigma(m)] = \frac{1}{1-y} + \frac{1}{1-\gamma y}.$$

(A.6)

Note that $F(y)^N$ acts as the partition function of the system when $N$ is fixed.

From $Z(z, y) = \frac{1}{1-zF(y)}$ one can calculate $\langle N \rangle = z\frac{d}{dz}\ln Z(z, y)$ and $\langle M \rangle = y\frac{d}{dy}\ln Z(z, y)$ and set $\langle N \rangle + \langle M \rangle$ to a desired value of $L$ to eliminate $z$. Particle density $\rho(y) = \frac{\langle N \rangle}{L}$ in GCE is then,

$$\rho(y) = \frac{1}{1 + y\frac{F'(y)}{F(y)}} = \frac{(1-y)(1-\gamma y)(2-y-\gamma y)}{(1-\gamma y)^2 + (1-y)^2}.$$

(A.7)

To verify if MPSS obtained here is indeed a good approximation let us calculate and compare from Monte Carlo simulations, the steady state values of $\eta_+$, the average number of beads per $+$ urn and $\rho_+$, the fraction of urns having internal degree $+$,

$$\eta_+ = \frac{1}{N}\sum_{k=1}^{N}\langle m_k \delta_{\sigma_k,+}\rangle; \qquad \rho_+ = \frac{1}{N}\sum_{k=1}^{N}\langle \delta_{\sigma_k,+}\rangle.$$

(A.8)

Since simulations are done at some specific $L, N$, we can use $F(y)^N$ as the partition function of the system; thus $p_+(m) = y^m/F(y)$ and $p_-(m) = \gamma^m y^m/F(y)$ and,

$$\eta_+ = \frac{1}{F(y)}\sum_{m=0}^{\infty} m Tr[X_+(m)]y^m = \frac{y(1-\gamma y)}{(1-y)(2-y-\gamma y)};$$

$$\rho_+ = \frac{1}{F(y)}\sum_{m=0}^{\infty} Tr[X_+(m)]y^m = \frac{1-\gamma y}{2-y-\gamma y}.$$

(A.9)

Using density-fugacity relation (A.7), both $\eta_+$ and $\rho_+$ can be obtained for different $\rho$.

In Fig. 3 we plot $\eta_+$ and $\rho_+$ as a function of $\gamma$ (dashed lines), for different $\rho$ in the range $(0.1, 0.9)$, along with those obtained from the Monte Carlo simulations of the model (solid lines). They match quite well for all $\gamma < 1$, indicating that, the approximate MPSS describes the RTP model very well.

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
