# Peer review of "Nonexistence of motility induced phase separation transition in one dimension"

_SciPost Physics, doi:SciPost Phys. 14, 165 (2023)_

## Round 1 · Referee Report · Anonymous (Referee 1) · 2022-10-18

Report

Dear Editor,

in this Letter, the authors present a mapping from a one-dimensional run-and-tumble dynamics with restricted tumbles to an urn model. They analyze the latter in the limit corresponding to two particles and show that, were this limit to faithfully describe the whole density range, phase separation would be impossible. They carry out a number of numerical simulations to support their claim that this limit is indeed qualitatively faithful.

The main appeal of the paper put forward by the authors is that questions have been raised "about the stability of [MIPS] states in 1D even though fluctuating hydrodynamic equations have predicted them [6,7]". References 6 & 7 correspond to early papers by Cates and Tailleur. However, Reference 6 states:

"In practice, of course, the very existence of a thermodynamic mapping in this (strictly 1D) system ensures that the bulk phase separation predicted by mean-field theory is replaced by Poisson-distributed alternating domains of mean size".

I.e., there is no phase separation in 1d. I thus think that, contrary to the authors' claim, phase separation in 1d has never been considered possible. Furthermore, the authors consider a model in which at most one particle per site is allowed. It has been shown by Soto and Golestanian that this only leads to finite-size clusters (in 1d and 2d; See ref 14 and [Sep\'ulveda Soto Phys. Rev. E 94, 022603 2016]). I thus think that the results presented by the authors are rather in line with the existing literature, contrary to what the article suggests.

That said, the mapping to the urn model is interesting, and reminiscent of the mapping between the TASEP and the ZRP (See Section 2 of [MR Evans and T Hanney 2005 J. Phys. A: Math. Gen. 38 R195). Unfortunately, its analysis is very hard to follow and the algebra presented on page 3 will be impossible to understand for readers who have not worked on matrix ansatz and related models.

For these two reasons, I do not think that the short format chosen by the authors is appropriate to convey their otherwise interesting results: the main message is not novel enough that a short format is required and the latter makes the article very hard to read. I think this paper would have much more impact if it was extended and all the results were detailed and explained.

Finally, I do not think that Scipost Physics is the right venue for this article. The journal is described as publishing articles which "provide details on groundbreaking results obtained in any (sub)specialization of the field". I do not think that the current submission fits this type of description: it is a particular case-study of a result that (I think) has been considered established since the early days of MIPS. That said, the results and the mapping are interesting and an extended, heavily improved, version could be considered for a more specialized publication.

More specific comments:

  • In the introduction, the authors write "Theoretical investigations of this phenomenon have thus far concentrated on continuum models [8-11]". I think that Ref 8 deals with lattice models and would be better placed below with ref 14. The reference [Whitelam, Klymko, Mandal, "Phase separation and large deviations of lattice active matter.", The Journal of chemical physics 148.15 (2018): 154902] should also probably be cited.

  • The second paragraph of the second column of the introduction should be fixed to reflect the content of the cited articles.

  • Speaking about grand canonical ensemble in the introduction is surprising at first since ensemble equivalence for active matter has not been demonstrated. I think this is coming too early and that it would be much clearer later on, once the factorized steady-state has been put forward.

  • "Nonexistence of MIPS transition in restricted tumbling model would imply that the same can not occur in any other RTP model in 1D". "The same" refers here to "Nonexistence of MIPS". I think the authors wanted to state the opposite: "Nonexistence of MIPS transition in restricted tumbling model would imply that MIPS cannot occur in any other RTP model in 1D".

  • The "restricted" tumbling hypothesis is a crucial difference between this model and other models previously studied. Its definition and its discussion should come much earlier in the article. The role of the asymmetry (right neighbor vs any neighbor) should also be discussed.

  • When the authors discuss the mapping to the beads-in-urn model, they should comment on the relationship with the mapping from TASEP to ZRP.

  • The beginning of the second column of page 2, when the current J(m) is discussed, is very hard to understand. This part should be clarified and detailed.

  • The two-urn section is not very clear: it implies $N=2$ but then an arbitrary $N$ comes back in Eq (2) and the N=2 case is only introduced later on in page 3. This should be completely rewritten: if the authors first want to do the general case, they should do so and only specify later on to the N=2 case. If they want to consider N=2 all along, they should do so. Furthermore, page 3 is amazingly dense and hard to read. I would strongly suggest detailing this part so that this can be understood by active-matter readers who are not familiar with the matrix-product ansatz.

  • The authors should discuss the implication of the larg- M limit, on page 3. This seems to imply a dilute limit which is not where MIPS is expected. Later on, they show the exponential distributions to survive at larger densities and I suggest that the authors clearly explain if this is a justification for using their results beyond the low-density limit.

  • Figure 2: The authors should explain how the color code is related to the values of $\omega$.

  • validity: -
  • significance: -
  • originality: -
  • clarity: -
  • formatting: -
  • grammar: -

Author:  Pradeep Kumar Mohanty  on 2022-11-24  [id 3070]

(in reply to Report 1 on 2022-10-18)

A reply to Referee 1 is attached as apdf file.

Attachment:

referee_1.pdf

---

## Round 1 · Referee Report · Anonymous (Referee 3) · 2022-10-20

Strengths

  1. The manuscript addresses an important physical question : whether or not certain run-and-tumble models can show phase separation in one dimension.
  2. The authors introduce a lattice model of restricted tumbling which is amenable to well established tools of analytical treatment.
  3. By mapping an interacting run-and-tumble model to one of beads-in-urn, the manuscript presents a useful technical path in studies of interacting one dimensional active systems.

Weaknesses

The paper, although well-written at parts, is still too technical in its approach. While it is a judicious choice by the authors to maintain the flow and not bombard the text with jargon, I also feel that some further explanation of the mathematics would really help the article.

Report

The authors map a system of interacting restricted run-and-tumble particles to a system of beads in urns. This mapping allows for an analysis of the original system through Matrix Product Ansatz, and shows that certain models of interacting run-and-tumble particles cannot produce a sustained phase separated state in the steady state. I believe the work is interesting and provides an alternate route (from the coarse-grained studies involving off-lattice velocity jump processes) to study interacting active matter physics in one-dimensional geometries. To my knowledge, the work is original and has the potential to lead to follow-up work by the community working in theoretical aspects of constrained active matter systems. However, I also have the following concerns with the text and I feel addressing these will only enhance the readability of the manuscript for a wider audience (beyond the community working in exact analysis of interacting systems).

  1. The analysis in page 3 can benefit from some physical explanations., for example, a term-by-term explanation of Eq. (3). 1a. Can we extract any physical insight from the choice of auxiliary matrices?

  2. Are the simulations for obtaining F_r, F_l and F performed in the steady state? Then are the two urns (or particles) chosen at random? Or do the authors choose specific initial states?

  3. What are the functional forms of \eta_{+} and \rho_{+} in terms of \gamma?

  4. Can the authors comment on some transient properties? For example, if we had an initial state of a given density with all particles being placed on consecutive lattice sites and the rest empty, i.e., we start with a macro-cluster. Is there a (heuristic) time-scale for this cluster to break down into a state of globally homogeneous density? I expect the transient states to depend on the hopping and tumbling rates, as well as on the density of particles. Ca this line of thought give any insight for the non-existence of MIPS in the model?

  5. The authors assume that the hop rate u(m_k, m_{k+1}) depends on the number of particles in the departure site k and the arrival site k + 1. Can they elaborate on the insight behind this assumption? Should the hop rate also not depend on the internal states of the arrival and departure urns? Naively, one might assume the current within a domain to depend on the polarities of the particles at the end of the domain.

  6. Can this restricted tumbling model show or rule-out micro-phase separation with formation of stable micro-clusters in the steady state?

  7. Indeed the manuscript could benefit from citing further recent literature on interacting one-dimensional run-and-tumble systems.

Requested changes

Kindly address the points mentioned in the report and correct typos, e.g., 'sate' -> state ,'dependn' -> depends .

  • validity: top
  • significance: good
  • originality: high
  • clarity: good
  • formatting: excellent
  • grammar: perfect

Author:  Pradeep Kumar Mohanty  on 2022-11-24  [id 3069]

(in reply to Report 3 on 2022-10-20)

A reply to Referee 3 is attached as a pdf file.

Attachment:

referee_3.pdf

---

## Round 1 · Referee Report · Anonymous (Referee 4) · 2022-10-28

Report

This is an interesting paper that addresses the question of whether motility-induced phase separation (MIPS) can occur in a simple model of run-and-tumble particles in one dimension. By mapping the dynamics to an urn model, coarse-graining and then solving with a matrix-product ansatz, the authors conclude that MIPS is not possible in this system. This corroborates certain existing findings, for example the work of Ref [14] which indicates that the domain size grows continuously as the tumble rate decreases, i.e., without any nonanalyticities of the type that might be associated with a phase transition. There may however be some regimes, such as that considered in Ref [27], where some kind of phase separation may occur, which are briefly discussed at the end of the manuscript.

I find this to be a timely contribution that address a question of current interest in the active matter / nonequilibrium statistical physics community. A particular strength is that it provides some analytical results for the many-body problem which, although not exact, go some way to improving our understanding of these driven systems.

Unfortunately, I found aspects of the presentation rather too terse for the manuscript to be easily followable, and I say this as someone who has worked extensively on 1d systems driven far from equilibrium. In particular the following sections of the manuscript would benefit from a considerable expansion to make the presentation digestible:

  1. It took me some time to realise that the third update in (1) allows only the left-hand particle to change direction ("tumble"), and then only when the right neighbour is moving the same way. I think it would help first of all to write out the $\omega_+$ and $\omega_-$ rules separately, and furthermore to state clearly in the text which updates are allowed and not allowed.

  2. Related to 1, I think some discussion of the rationale for this set of rules is required. I understand that this allows analytical progress later in the manuscript, but did not follow at what point this becomes important. Furthermore, one may worry that the rule breaks CP symmetry for those cases where $r_+=r_-$ for each of the transition rates $r\in{p,q,\omega}$ and such symmetry might be expected. I understand that setting certain tumbling rates to zero moves in the direction of making the system more likely to phase separate, so I doubt this affects the final conclusion, but one does feel nervous about removing a symmetry from a model that might initially be present.

  3. I did not follow the set-up of the coarse-grained model at all. The exact urn model has particles hopping in both directions, whereas the coarse-grained model seems to have them moving only to the right (or perhaps this is a misleading impression given by Fig 1c)? I did not see how the hop rates $u(m)$ are defined, whether these depend only on the occupancy of one site (as suggested by Fig 1c) or two sites (as suggested by the text). I really could not understand why some of the domains of size $m$ in Fig 1b overlap, or what determines the start and end of a domain. This entire mapping needs to be laid out completely and methodically.

  4. In Fig 2, where hop rates are obtained from numerical simulations, it was unclear what was being simulated here. Is it the full dynamics of the original model, or one of the derivatives described in the main text.

  5. I would like to see a bit more discussion of the comparison with Ref 27. Although the latter relies on a specific limit, I understood that the current analysis applied to any parameter combination so one ought be able to take the same scaling in the present work to compare more directly. Moreover, the $1/L$ separation of the run and tumbling rates I think corresponds to the scaling limit considered in Ref 15, which makes me wonder if something different happens in this scaling limit and whether this can be seen from the present analysis.

I have recorded these as "minor revisions" as I think all can be adequately dealt with by the authors simply expanding the text with information that should be easily accessible to them (as opposed to conducting further research). However, I think the first four are essential for a proper understanding of the manuscript.

  • validity: -
  • significance: -
  • originality: -
  • clarity: -
  • formatting: -
  • grammar: -

Author:  Pradeep Kumar Mohanty  on 2022-11-24  [id 3067]

(in reply to Report 4 on 2022-10-28)

A reply is attached as a pdf file.

Attachment:

referee_4.pdf

---

## Round 2 · Referee Report · Anonymous (Referee 4) · 2022-12-19

Report

The authors have made several revision to the manuscript in response to the reviewer comments (which were largely consistent between reviewers). Overall these have had the effect of clarifying the manuscript somewhat, and although the reader is still required to invest some effort in unpacking the mappings between the different models investigated, all the information one needs is there.

One of my questions in the first report however remains unresolved. This pertains to the presence, or otherwise, of CP symmetry in the dynamics. I agree with the authors that CP means: flip the spins, and reverse their order on the lattice. Under this transformation, configurations like $++$ turn into $--$ whilst $+-$ and $-+$ do not change (since spin flip is the same as order reversal here).

I agree that when $p_{\pm}=q_{-+}$ the particle hop dynamics exhibit CP symmetry. But, unless I am grossly misreading either the model dynamics or the notion of CP symmetry, the tumble dynamics do not. Consider for example the move $++ \to -+$ that takes place at rate $\omega$. Under CP transformation, this becomes $-- \to -+$$ which is not included in Eq (2) in the text. Therefore I do not believe that the tumble dynamics satisfy this symmetry. Actually, one can probably understand this already from the statement that tumbling is assisted from the right: under the P transformation right turns into left.

So I guess there are three questions.

(1) Does this model break CP symmetry or not?
(2) If so, does this matter? (Previously I suggested not, because it's the restriction that is important to the condensation argument, not symmetry; but nevertheless I feel that symmetry breaking is not something one does lightly)
(3) How does flipping being assisted from the right facilitate the mappings described in the main text? Does the mapping to the urn model depend on it?

Requested changes

Please address the point re CP symmetry set out above.

  • validity: -
  • significance: -
  • originality: -
  • clarity: -
  • formatting: -
  • grammar: -

Author:  Pradeep Kumar Mohanty  on 2023-02-02  [id 3300]

(in reply to Report 1 on 2022-12-19)

We thank Referee 1 for recommending publication of this article in SciPost. The question regarding symmetry is clarified in the article. An alternative model, with same run dynamics and tumbling occurring only when particles are assisted from left, leads to the same steady state as our model where tumbling occurs when particles are assisted from right (this can be seen easily by mapping both models to corresponding Urn models). Thus, although dynamics of our model violates CP symmetry (run-dynamics obeys it whereas tumbling does not), the steady state remains invariant under the CP transformation.

---

## Round 2 · Referee Report · Anonymous (Referee 3) · 2023-1-26

Report

I thank the authors for their effort in re-writing the manuscript and also for detailed responses to my comments.

The current manuscript is a much improved version of its predecessor. The presentation, as it stands now, is much easier to follow. Transporting the technical discussion to the Appendix has also helped the manuscript. (There are still various typos in the text, e.g., sate (state) or loose (lose), etc.)

The discussion in the current version makes clear several important things, such as why in the coarse-grained description one can neglect internal degree of freedom of the urns; or how the simulations were performed to infer about u(m).

Largely, I am in agreement with the subject matter of the text but I also feel that an important point was raised by one of the Referees regarding symmetry-breaking in the model.
Let's consider the model where a tumbling can occur only if the particle has neighbors on both sides (instead of one side). It seems that for such a situation there can still be a mapping to a beads-in-urn system : (a) the number of beads in each urn is now equal to the total number of vacancies on both sides of the particle, (b) leftwards hopping of the particle (corresponding to urn_k) to a vacant site is equivalent to a bead hopping from urn_(k-1) to urn_(k+1), (c) total number of beads is conserved and equal to half the sum of beads in all urns. Indeed, in the long-time limit any cluster can break as none of the four rates p+, p-, q+, q- are zero but whether or not the system becomes homogeneous for any density $0 <\rho < 1$ might be an open question.

Nevertheless, as I said previously, the manuscript has strong potential to lead to further work on both steady-state and transient problems in one dimension. Analysis of hardcore active systems at a microscopic level is not an easy task. I find the article to be a welcome contribution in this regard. Hence, I recommend publication.
  • validity: top
  • significance: high
  • originality: high
  • clarity: high
  • formatting: excellent
  • grammar: -

Author:  Pradeep Kumar Mohanty  on 2023-02-02  [id 3301]

(in reply to Report 2 on 2023-01-26)

We thank Referee 2 for recommending publication of this article in SciPost. In the revised version we have addressed all your comments.

Symmetry: An alternative model, with same run dynamics and tumbling occurring only when particles are assisted from left, leads to the same steady state as our model where tumbling occurs when particles are assisted from right (this can be seen easily by mapping both models to corresponding Urn models). Thus, although dynamics of our model violates CP symmetry (run-dynamics obeys it whereas tumbling does not), the steady state remains invariant under the CP transformation.

Symmetric tumbling: A model with same run dynamics and tumbling occurring only when particles are assisted from both left and right is being studied. We find exact steady states for four RTPs on a lattice. One can explicitly show that MIPS transition is absent there (work is in progress).

---

## Round 2 · Referee Report · Anonymous (Referee 1) · 2023-1-27

Report

Dear Editor,

The authors have improved their manuscript to address many of the
comments made by the referees. That said, the reading remains
difficult, with many statements that can easily be
confusing. Furthermore, I think the results do not match Scipost
editorial policy which aims at publishing articles which "provide
details on groundbreaking results obtained in any (sub)specialization
of the field".

Overall, this is an interesting technical contribution that maps a
slightly peculiar model of run-and-tumble particles onto an urn model
and then employs an approximation to rule out phase separation in
1d. I list below several parts which I still find confusing and should
probably be improved prior to publication. Once this is done, I would
support publishing this article in a specialized journal. Whether
Scipost Physics wants to do so is an editorial choice---I have no
strong opinion on that matter.

-/ Restricted tumbling. Particles can tumble only when the site on
their right is occupied. This should be stated explicitly in the
abstract. The current presentation is confusing.

-/ "In this article, we argue and show explicitly using 1D lattice
models of RTP that indeed MIPS can not occur in 1D". I think this
statement, which is repeated several times, is too strong. The
authors show this for a particular model. I agree this has
consequences beyond the sole model studied here but the authors
themselves discuss a counter example when the rates scale with the
system size. I do not think such generic and vague statements are
particularly useful. (The same goes for "a proof of nonexistence of
MIPS in our model necessarily guarantees its nonexistence in any
other model that has more liberal tumbling dynamics.", etc.)

-/ The authors speak about a "left <-> right" symmetry on page 2. I am
not sure to what they refer but their restricted tumbling rule breaks
the symmetry i -> L-i. Their model is thus not endowed with what I
would naively call a "Left-right symmetry".

-/ Please define $m_k$ in the caption of Figure 1. It is not possible
to understand it the first time it is cited without this definition.

-/ The authors mention several times that their coarse-graining
amounts to "averaging" the current over "internal degrees of
freedom". It's not clear what measure they use to do so
(flat?). Please write a clear mathematical definition of the
coarse-graining procedure. What is done is really not clear at the
moment (the second column of page 2 should be much clearer given the
importance of this coarse-graining/approximation).

-/ Equation (3) and above: is there a comma missing between \sigma_k
and m_k? This notation is not clear to me.

-/ Equation (6). Can the normalization Q be computed?

-/ Right after Equation (9). I suggest using a more careful
phrasing. All the computations so far are for N=2 particles. It has
been said several time in the literature that two-body interactions
are not sufficient to account for MIPS (see, e.g.,
[Phys. Rev. Lett. 126, 038002 (2021)]). Have the authors ruled out
more than that at this stage?

-/ Fig 2a: in the text, $w$ is said to vary from 0.03 to 10. In the
caption, it is said to vary from 0.05 to 10. Please clarify. Also,
the authors should use a color code allowing the reader to associate
the curves to the corresponding values of $w$.
  • validity: good
  • significance: ok
  • originality: good
  • clarity: poor
  • formatting: good
  • grammar: acceptable

Author:  Pradeep Kumar Mohanty  on 2023-02-02  [id 3302]

(in reply to Report 3 on 2023-01-27)

We thank Referee 3 for his/her useful comments and suggestions. We have incorporated all the suggestions in the revised manuscript.Those requires additional explanations/reply are mentioned below.

Counter example (MIPS in 1D ): In the counter examples, the tumbling rates are downplayed by a factor
inversely proportional to the system size, which vanishes in the thermodynamics limit. Exsitence of MIPS is
not surprising here as in absence tumbling, the system trivially goes to a jammed/absorbing-state.

Symmetry: An alternative model, with same run dynamics and tumbling occurring only when particles are assited from left, leads to same steady state as our model where tumbling occurs when particles are assited from right (this can be seen easily by mapping both models to corresponding Urn models). Thus, although dynamics of our model violates CP symmetry (run-dynamics obeys it whereas tumbling does not), the steady state remains invariant under the CP transformation.

Coarse-graining: The word coarse-graining is used here to denote 'an approximate macroscopic description'; it is not a strict mathematical procedure, one like, coarse-graining of lattice dynamics to obtain a effective continuum description.

Normalization Q: Explicit form of Q is mentioned just above Eq. (5).

---

## Round 2 · Author Response

Dear Editor,
Thank you for providing us an opportunity to resubmit the article to SciPost. We also thank all the referees for their valuable comments and suggestions. In the revised version we have incorporated all their suggestions. The article has gone through a major revision, with restructuring of the texts and clarification of the issues raised by the referees.

Separately, we have prepared replies to all the referees, addressing their queries point by point. Replies to all three referees are uploaded to SciPost (as pdf files) just below the respective referee comments. We have also added the summary of changes there.

We sincerely hope that you will find the revised article suitable for publication in SciPost.

Thanking you again for your kind consideration.
With regards,
Indranil Mukherjee, Adarsh Raghu, and P. K. Mohanty

---

## Round 2 · List of Changes

Summary of changes

The article has gone through a "major revision" following the valuable comments of the referees. It is difficult to

provide details as in the revised manuscript, the whole structure and almost all the paragraphs are modified.

Only some important changes are listed below.

  1. Restricted tumbling dynamics is now given separately as Eq. (2) followed by a longer discussion.

  2. Fig. 1 is modified - exact and coarse-grained urn models are now described more clearly in Fig1(b).

  3. Discussions on Matrix Product Ansatz (MPA) is described in the APPENDIX. We hope it helps the readers to arrive at the results and conclusions of the article without bothering much about the detailed mathematical steps of MPA.

  4. New references [15], [16] and [23] are added.

---

## Round 3 · Author Response

Dear Editor,
Thank you for giving us an opportunity to resubmit the article - we have revised the manuscript following the suggestions by all the referees . But somehow, We did not find a link to resubmit the revised manuscript. Kindly look into the matter. Please note that we have posted replies to all the referees.

Thanking you again,
with regards,
Indranil Mukherjee, Adarsh Raghu and P. K. Mohanty

---

## Round 3 · List of Changes

In the revised manuscript we have addressed all the comments and suggestions made by three referees. In the pdf file the changes are marked in red.

---

## Editorial Decision

published